# Synthesis and characterization of cerium doped $N_iZ_n$ nano ferrites as substrate material for multi band MIMO antenna

**Raees Muhammad Asif** [1]*, **Abdul Aziz**[1], **Majid Niaz Akhtar**[2], **Muhammad Azhar Khan**[2], **Muhammad Nawaz Abbasi**[1], **Hafiz Abdul Muqeet**[3]

**1** Faculty of Engineering & Technology, The Islamia University of Bahawalpur, Bahawalpur, Pakistan, **2** Institute of Physics, The Islamia University of Bahawalpur, Bahawalpur, Pakistan, **3** Dept. of Electrical Engineering and Technology, Punjab Tianjin University of Technology, Lahore, Pakistan

* raees.asif@ptut.edu.pk

**Data Availability Statement:** All relevant data are within the manuscript and its Supporting information files.

## Abstract

In addressing issues related to electromagnetic interference, the demand for ferrite materials with exceptional magnetic and dielectric properties has escalated recently. In this research, sol-gel auto combustion technique prepared Nickel zinc ferrites substituted with cerium, denoted as $Ni_{0.5}Zn_{0.5}Ce_{0.02}Fe_{1.98}O_4$.X-ray diffraction (XRD), Vibrating Sample Magnetometer (VSM), and Field Emissions Scanning Electron Microscope (FESEM) were used to investigate the structure, magnetic properties, and morphology of Cerium doped $N_iZ_n$ Nano ferrites, respectively. The magnetic and dielectric properties of the sample was examined within a frequency range of 2.5–5.5 GHz. Sample exhibits low permittivity (2.2), high permeability (1.4), low dielectric (0.35) and magnetic loss tangent (-0.5) and highest saturation magnetization measuring 30.28 emu/g. A Novel Double-band, 4x4 MIMO window grill-modeled antennas operating on 3.5 GHz and 4.8 GHz frequency bands for 5G smartphones is designed using the CST microwave studio suite. The performance of window grilled 4x4 MIMO antenna model with Cerium doped $N_iZ_n$ nano ferrites as substrate, is investigated and found the return loss of -35 and -32 dB, with the bandwidth of 200MHz, gain (1.89 & 4.38dBi), envelope correlation coefficient (0.00185), channel capacity loss (0.2bps/Hz), and interterminal isolation of (22& 19dB).The results show that the antenna size is reduced with improved bandwidth, higher isolation and better diversity gain performance using Cerium doped $N_iZ_n$ nano ferrite substrate compared to conventional dielectric substrates.

## Introduction

MIMO (Multiple-Input Multiple-Output) is an antenna technology for wireless communications in which multiple antennas are used at both the source (transmitter) and at the destination (receiver). MIMO technology minimizes errors, optimize data speed and improve the capacity of radio transmissions by enabling data to travel over multiple signal paths at the same time [1]. Small, lightweight MIMO antennas are crucial in achieving these goals. However, developing

**Funding:** The author(s) received no specific funding for this work.

**Competing interests:** The authors have declared that no competing interests exist.

MIMO antennas is a challenge as their size, weight, and separation between nearby antennas can significantly impact the radiation performance of overall MIMO system [2]. Various methods have been developed to improve the data transfer capacity, separation between nearby antennas, and reduce the size of MIMO antenna components. However, some of these methods may negatively affect the radiation performance of the antenna [3]. A suitable magnetodielectric material can be used as a substrate, which can help achieve the necessary performance parameters like a miniaturized antenna with a high data transfer rate without affecting the radiation performance of the antenna. Several studies have shown the effectiveness of using a magnetodielectric substrate in improving the performance of MIMO antennas [4]. In recent years, ferrite materials with extraordinary qualities have been essential for overcoming the problems posed by contemporary electromagnetic interference. These attributes include high reflectivity and retention capacity, lightweight design, and tiny size.

Bhongale, S.R., discovered that a Mg-Nd-Cd ferrite substrate resulted in a smaller antenna size and better radiation performance than a standard dielectric substrate [5]. Ferrite material has been used as the substrate for millimeter wave applications, the antenna size was reduced, and bandwidth increased [6–9]. In the case of the X-band, using nickel ferrite as a substrate, decreases antenna dimensions and increases bandwidth [10]. It was reported in the literature that employing $Ni_{0.6}Zn_{0.2}Co_{0.2}Fe_{1.98}O_4$ as an antenna substrate reduces antenna scaling and magnetic losses [11]. However, a Dy-Sm doped Mg-ferrite substrate decreased antenna properties due to the ferrite's high dielectric constant [12]. A 2x2 MIMO antenna on a ferrite substrate resulted in a lower gain of -8.83 dBi [13]. R M. Asif, A. Aziz reported that a Terbium-doped $Ni Zn$ ferrite substrate resulted in a smaller antenna size and better radiation performance in a 4×4 dual-band MIMO antenna than a dielectric substrate [14].

$Ni Zn$ ferrites have been extensively used in high-frequency scenarios because of their strong saturation magnetization, enormous electrical resistivity, lesser dielectric and magnetic losses, low coercivity, and eddy current losses. However, it is observed that doping rare earth ions can further modify the electromagnetic characteristics of $Ni Zn$ ferrites. Rare-earth ions have emerged as promising additions for enhancing the magnetic properties of ferrites. Among the various rare earth metals, Cerium has played a crucial role in improving the characteristics of spinel ferrites [15–17].

Cerium is a soft, silvery-white metal abundantly found in the Earth's crust. It exhibits excellent electrical conductivity and possesses elevated melting and boiling points [18]. In a recent development, [19] cobalt-zinc ferrites doped with cerium demonstrated significant absorption of the electromagnetic spectrum in the near-infrared region [20]. Suggested that cerium-doped cobalt ferrites have high potential for use in switching memory devices. Additionally, [21] investigated the structural, optical, and magnetic properties of cerium-doped $Ni Zn$ ferrites and concluded that the proposed sample is suitable for high-frequency applications.

Based on these results, the current study aims to synthesize Cerium-doped $Ni Zn$ ferrite and utilize it as a substrate in the design of MIMO antennas.

Several benefits were achieved by incorporating Cerium doped $Ni Zn$ spinel ferrite as the substrate material in the dual-band MIMO antenna in this work. These advantages encompass an increased bandwidth of 200 MHz in each frequency band, enhanced isolation of 23 dB between the dual-band 4x4 MIMO antenna elements, and high gain of 4.2 dBi.

## Experimental works

### Synthesis and characterization of cerium doped $Ni Zn$ ferrite

For the initial experiment, nanocrystalline ferrite particles with the composition $Ni_{0.5}Zn_{0.5-}Ce_{0.02}Fe_{1.98}O_4$, were synthesized using the sol-gel auto-combustion technique. The production

conditions were kept identical to enable comparative analysis. Stoichiometric proportions of all metal nitrates such as, $N_i(NO_3)_2 \cdot 6H_2O$, $Z_n(NO_3)_2 \cdot 6H_2O$, $C_e(NO_3)_3 \cdot 9H_2O$ and $F_e(NO_3)_3 \cdot 9H_2O$ were calculated as initial precursors. All these initial precursor salts for the composition under study were purchased from Sigma Aldrich Company in the powder form and were of high analytical purity.

The sol-gel auto-ignition approach is favored among multiple ferrite preparation techniques due to its adequate control over stoichiometry, quick and efficient generation of nano-particles, and low cost [15]. Equimolar quantities of citric acid ($C_6H_8O_7$) and nitrates were mixed individually with 100 milliliters of purified water that swirled for 24 hours to create a homogeneous mixture. The resultant solutions were mixed, and the pH was raised to 7 by adding an ammonia solution while the mixture was continually agitated. The solution was lit and burned after being heated to 80°C and agitated until it formed a dry gel. The sol-gel precursor solution is first heated to form a gel, which is then ignited using the fuel source. The combustion of the fuel generates a high temperature, which causes the gel to undergo a rapid exothermic reaction and form a solid material [22]. After that, the powder was allowed to naturally cool to room temperature. The morphological and structural characteristics of the powder samples were assessed upon sintering. The sample synthesized was sintered at 700°C in a furnace for 7 hours for maximum crystallinity.

The sintered powder ferrite sample was pressed into pallets under 25MPa pressure using a uniaxial hydraulic press. Disc shaped pellets with diameter of 14 mm and thickness of about 3 mm were obtained for dielectric measurements. In order to remove any residual moisture and densify, pallets were again sintered in an oven at a temperature of 500°C for 4 hours. These pellets were used for measuring dielectric parameters on VNA.

Synthesized material was characterized through the XRD method with a Bruker D2-Phaser instrument. Using Cu-K radiation with a two-theta range of 10–80°, the X-ray diffraction (XRD) patterns of the samples were investigated at room temperature. The composite material's shape and the size of grains were assessed using a Nova FEI Nano SEM 450. This scanning electron microscope has field-emission technology and can achieve a maximum resolution of 1.8 nm. It operates at a voltage of 03 kV. The magnetic properties of the specimen are evaluated at ambient temperatures using a vibrating sample magnetometer. Vector Network Analyzer (Nano VNA-F) examined the magnetic and electrical characteristics.

## Findings and analysis

### XRD

The XRD technique presented in Fig 1 efficiently confirms the phase of the Cerium-doped $N_iZ_n$ ferrite. The observed broad peaks in the XRD pattern provide evidence of fine particles with the ferrite sample's single-phase, cubic crystal structure. Moreover, the specific planes of (220), (311), (400), (333) and (440) are observed. Mean crystallite size is also examined using Debye Sherrer's formula from the (440), (311), and (220) peaks, resulting in a size of 50 nm. This validates the nanocrystalline nature of the ferrite sample.

$$D = K\lambda / w \cos\theta, \tag{1}$$

where D is the average crystallite size (in nanometers), K is the shape factor (typically assumed to be 0.9) θ is Bragg's reflection angle, 'w' represents the full-width at half maximum (FWHM) of the diffraction peak (in radians) while 'λ' denotes the wavelength of the X-rays, which is 0.154 nm.

$$a = d_{hkl}(h^2 + k^2 + l^2)^{\frac{1}{2}} \tag{2}$$

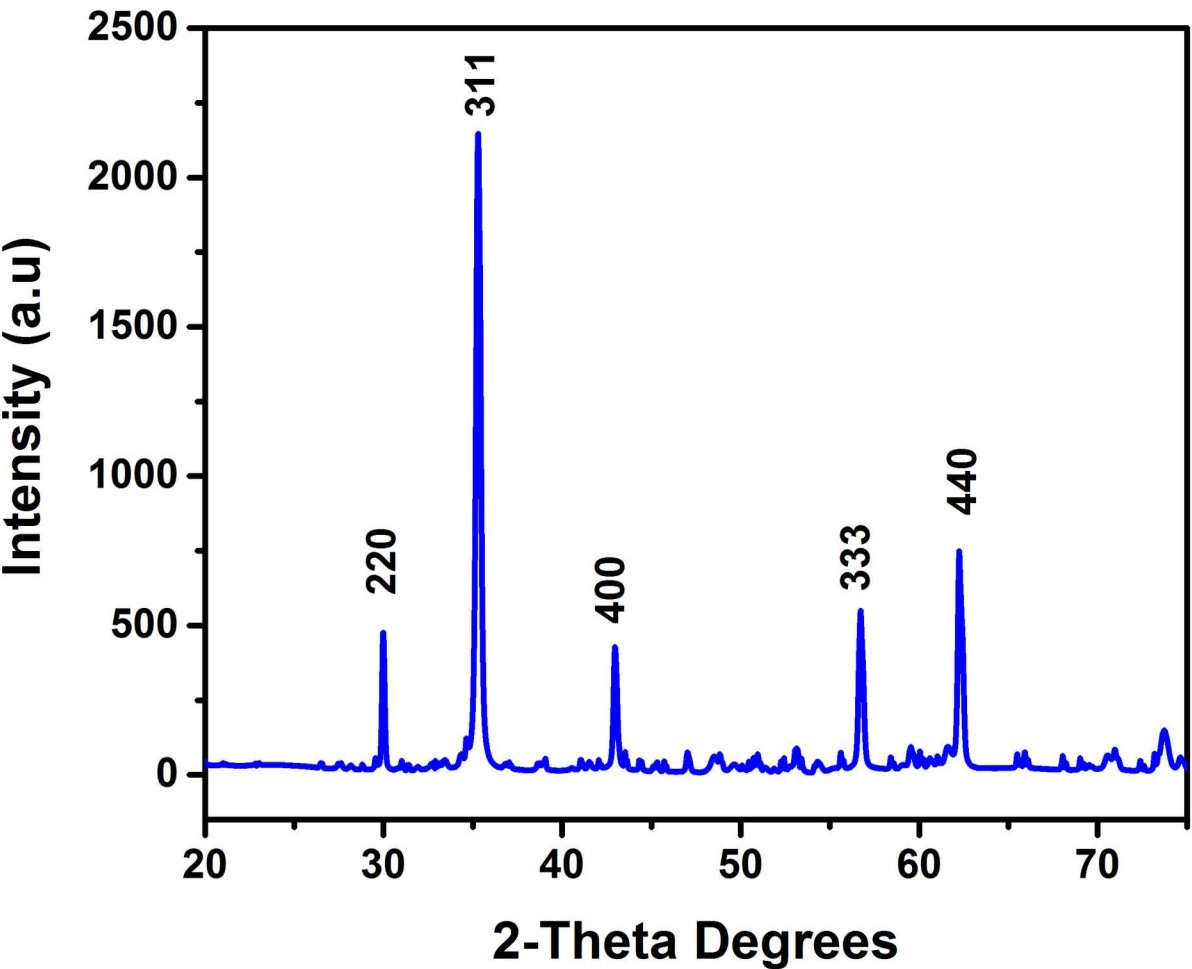

**Fig 1. The XRD—X-ray diffraction graph, with the chemical composition of $Ni_{0.5}Zn_{0.5}Ce_{0.02}Fe_{1.98}O_4$ ferrite produced by sol-gel auto ignition procedure.**

The lattice parameter "a" is determined to be 8.34Å by employing the above-mentioned equation. Table 1 shows the values of FWHM, d-spacing, lattice parameter, bulk density, X-ray density, % porosity, height (cents), micro strain, cell volume, and crystalline size.

Microstrain is calculated from XRD data using Origin software and by using the relation $\varepsilon = \frac{\beta}{4\tan\theta}$. Where $\varepsilon$ is the microstrain in radians, which is the ratio of XRD peak width to the peak position. $\beta$ is the line broadening at full width half maximum (FHWM) in radians and $\theta$ is the Bragg's angle in degrees.

**Table 1. XRD parameters of cerium doped $N_iZ_n$ ferrite.**

| Sample | d-Spacing (Å) | Lattice Parameter (Å) | Bulk Density | X-Ray Density | % Porosity | Height (Cents) | FWHM | Crystallite Size | Cell Volume (Å) | Micro Strain (%) |
|---|---|---|---|---|---|---|---|---|---|---|
| NiZn-Ce | 2.518 | 8.34 | 3.25 | 3.95 | 17 | 140.8 | 0.294 | 35 | 583.5 | 0.0018 |

Archimedes principle was used to measure the mass and volume of sample under investigation and then the density with the following relation,

$$\rho_{sample} = \frac{m_{sample}}{v_{sample}}$$

Similarly, the porosity which is the measure of the void spaces within the material obtained by using the following relation $\emptyset = \frac{v_{void}}{v_{sample}}$ Where, $v_{sample}$ is the total volume of the sample, $v_{void}$ is the volume of the void spaces within the sample.

## SEM images of cerium doped $N_iZ_n$ ferrite

Fig 2 shows a homogeneous dispersion of aggregate-forming nanoparticles during the fusion and ignition process. These aggregates are porous, and the SEM image makes this very evident. The grains that make up the aggregated crystallites are closely linked and have a cuboid form. Due to the compaction and grain growth during the sintering process at a temperature of 700°C, a significant aggregation of numerous crystallites has occurred, leading to an average grain size ($G_{avg.}$) of between 65 and 86 nm, calculated using the linear intercept technique with the following equation.

$$G_{avg.} = {}^{1.5L}/_{MN} \tag{3}$$

Whereas, $G_{avg.}$ = Average grain size, N = Number of intercepts, M = magnification, L = Specimen line length.

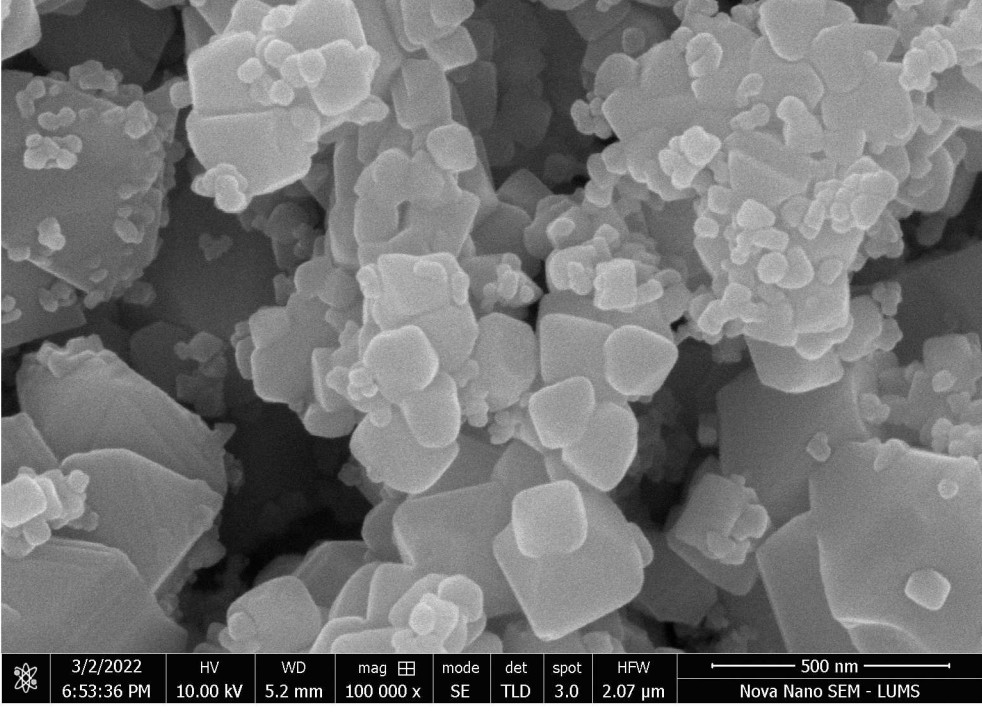

**Fig 2. SEM microphotograph for $Ni_{0.5}Zn_{0.5}Ce_{00.2}Fe_{1.98}O_4$ ferrite.**

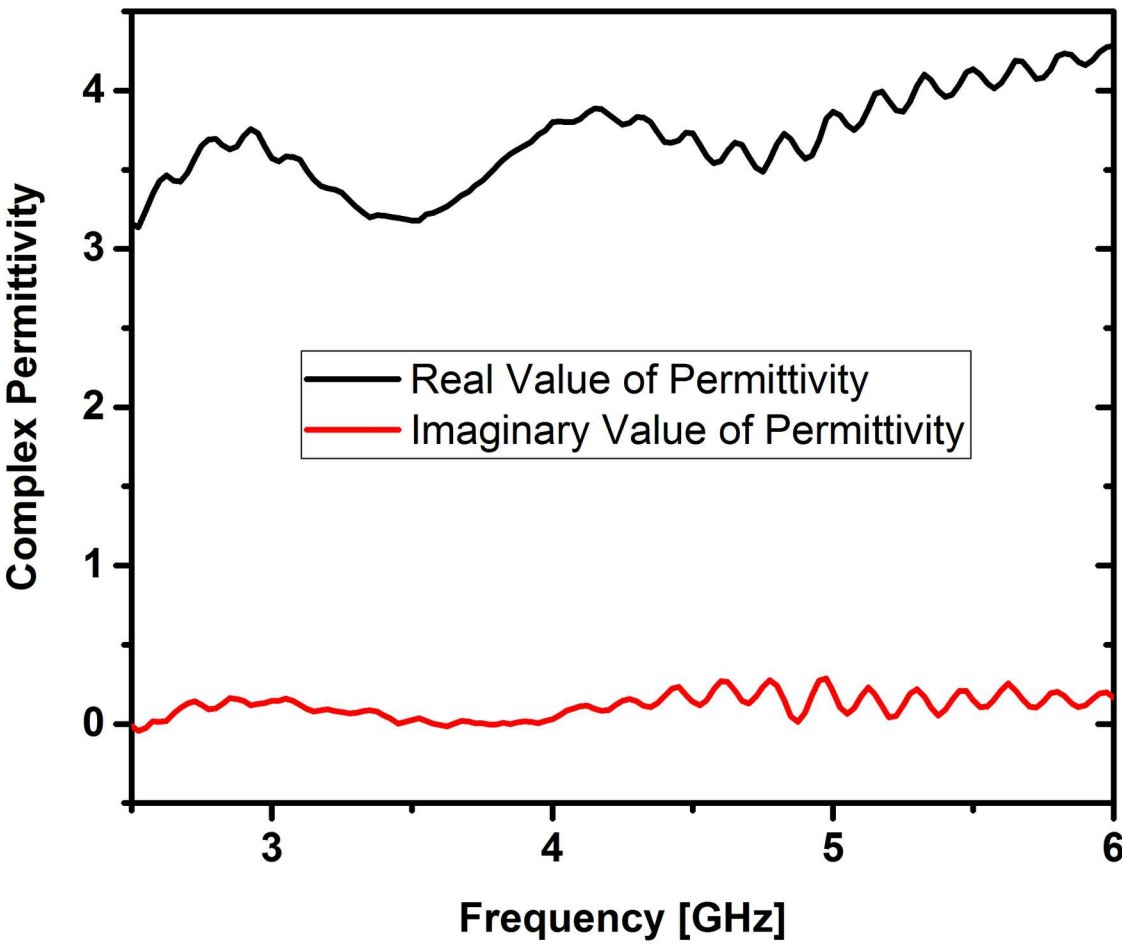

**Fig 3. Complex permittivity of cerium doped $N_iZ_n$ ferrite annealed at 700˚C.**

### Dielectric constant and dissipation factor measurements

The behavior of ferrites in terms of dielectric properties is closely linked to their conduction mechanism [23, 24]. Dielectric and magnetic properties of Cerium doped $N_iZ_n$ ferrites were obtained with the help of an"Epsilometer with R60VNA". Complex values of permittivity and permeability were obtained by using the equations $\varepsilon_r = \varepsilon_r' - j\varepsilon_r''$ and permeability $\mu_r = \mu_r' - \mu_r''$ after post-processing of S-parameters and are shown in Figs 3 and 4. The real part of permittivity or dielectric constant ($\varepsilon'$) and permeability ($\mu'$) relates to the ability to store electric and magnetic energy within the material from an external electrical and magnetic field, respectively. Also, the imaginary part of permittivity ($\varepsilon''$) and permeability ($\mu''$) corresponds to the loss (dissipation) of electric and magnetic energy due to the external electrical and magnetic field, respectively. The following equations were used to calculate the dielectric and magnetic losses $\tan \delta_e = \frac{\varepsilon''}{\varepsilon'}$ and $\tan \delta_m = \frac{\mu''}{\mu'}$.

The reason for the lower dielectric loss values obtained can be linked to the reduction of $Fe^{2+}$ ions during the sol-gel process. This reduction leads to enhanced crystal arrangement and molecular stoichiometry [25]. Minimizing the losses of prepared Nano ferrite samples is important as this affects the efficiency. It is widely recognized that the characteristics of initial permeability are influenced not only by the chemical composition but also by the sintered

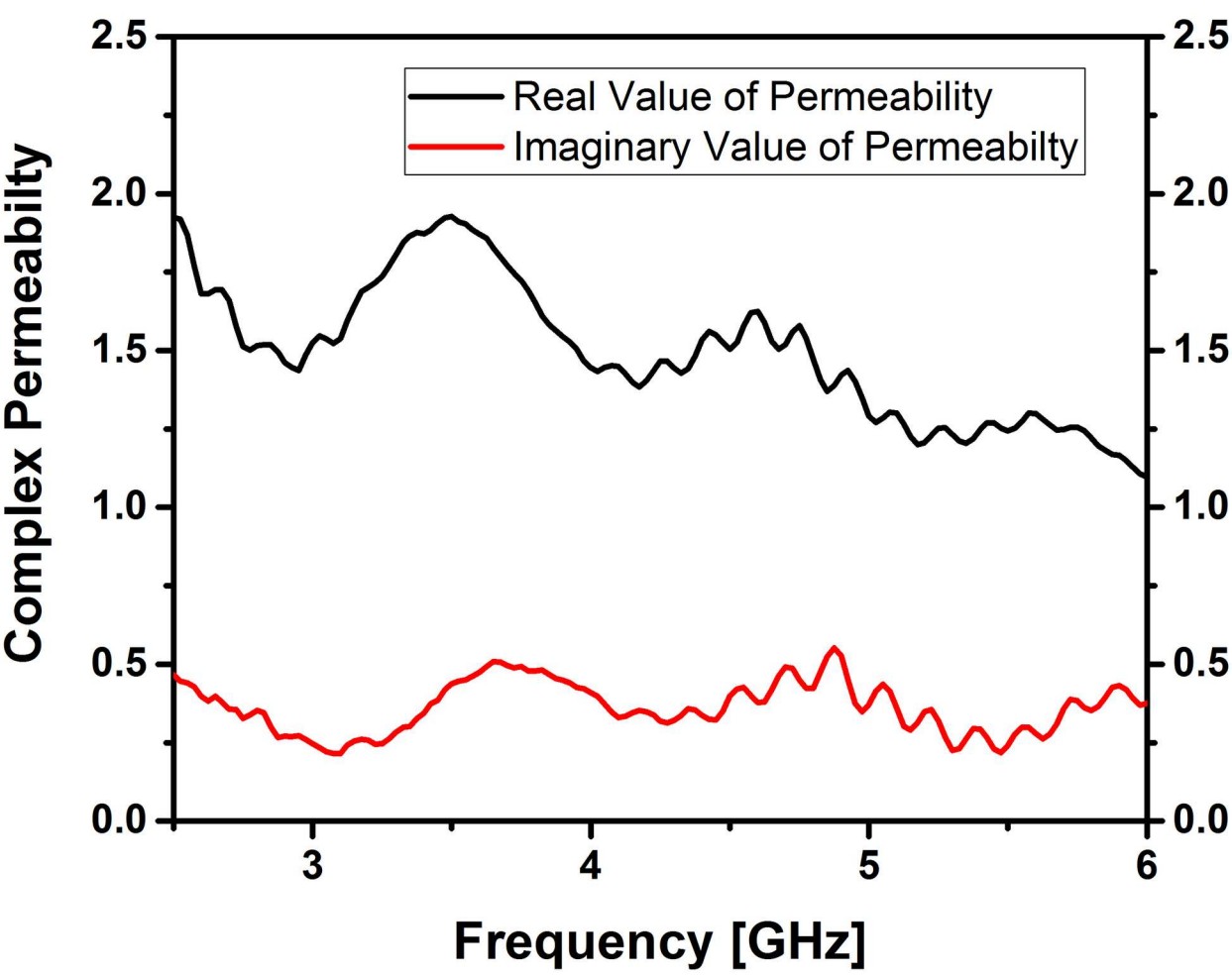

**Fig 4. Complex permeability of cerium doped $N_iZ_n$ ferrite annealed at 700˚C.**

body. By controlling the microstructures, it becomes possible to achieve the desired magnetic properties. Additionally, the samples $Ni_{0.5}Zn_{0.5}Ce_{0.02}Fe_{1.98}O_4$ in this study also adhere to the *Snoek's* Law [26, 27].

$$(\mu_s - 1)f_r = \frac{2}{3}(\gamma . M_s) \tag{4}$$

Where $\mu_s$ is the static permeability, $M_s$ is the saturation magnetization and $\gamma$ is the gyromagnetic ratio. Antenna miniaturization factor (n) can be determined using the relation $\lambda_{eff} = \lambda_\circ / \sqrt{\mu_r \varepsilon_r}$ Where, $\lambda_\circ$ is the free space wavelength.Here optimal values of permittivity ($\varepsilon_r$) and permeability ($\mu_r$) values of the fabricated ferrite substrate contributed to minimizing antenna dimensions while enhancing bandwidth characteristics.

These characteristics are crucial when employing the material as a substrate in MIMO antennas. The measurements indicate that the mean dielectric constant is 2.2, and the mean magnetic constant is 1.4 for the ferrite sample, as illustrated in Figs 3 and 4. These intriguing parameters suggest that the Cerium doped $N_iZ_n$ ferrite, with the chemical composition $Ni_{0.5}Zn_{0.5}Ce_{0.02}$-$Fe_{1.98}O_4$, holds promise as a substrate for antennas in high-frequency applications.

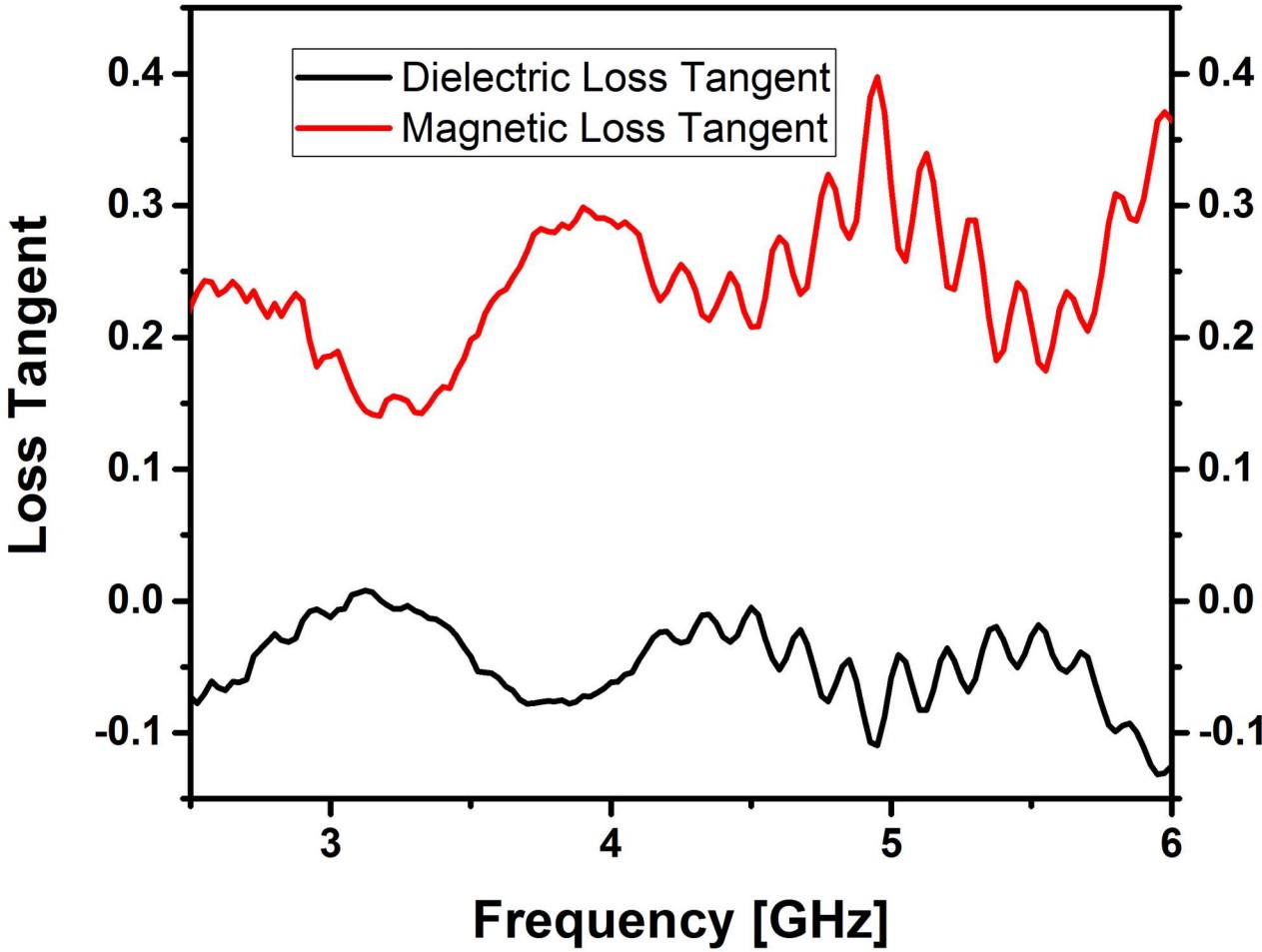

**Fig 5. Variation of dielectric loss tangent (tan δ$_\varepsilon$) and magnetic loss tangent (tan δ$_\mu$) with frequency for cerium doped $N_iZ_n$ ferrite annealed at 700˚C.**

Fig 5 reveals that the mean tan δ$_\varepsilon$ is 0.35 and the mean tan δ$_\mu$ is -0.5 for this material; this signifies a considerably lower dissipation factor, indicating reduced losses. Negative values of tan δ indicate that the inductive behavior (μ') dominates over the resistive behavior (μ") of the sample under investigation at the given frequency range and magnetic field strength. This implies that the material exhibits more magnetic storage (inductive) properties than magnetic dissipation (resistive) properties and may be due to complex interactions between magnetic domains, hysteresis effects and domain wall movements.

### Magnetic hysteresis

The magnetic hysteresis loop of the Cerium doped $N_iZ_n$ ferrite material is illustrated in Fig 6 and is obtained by using a vibrating sample magnetometer. This confirms that the sample is ferromagnetic. Magnetic properties of the ferrite sample are used to evaluate the sample's high-frequency response. The Magnetic parameters of the synthesized ferrite sample are listed in Table 2. The ferrite sample's operating microwave frequency is calculated using the following relation and is directly related to the device's performance [28].

$$\omega_m = 8\pi^2 M_s \times \gamma \tag{5}$$

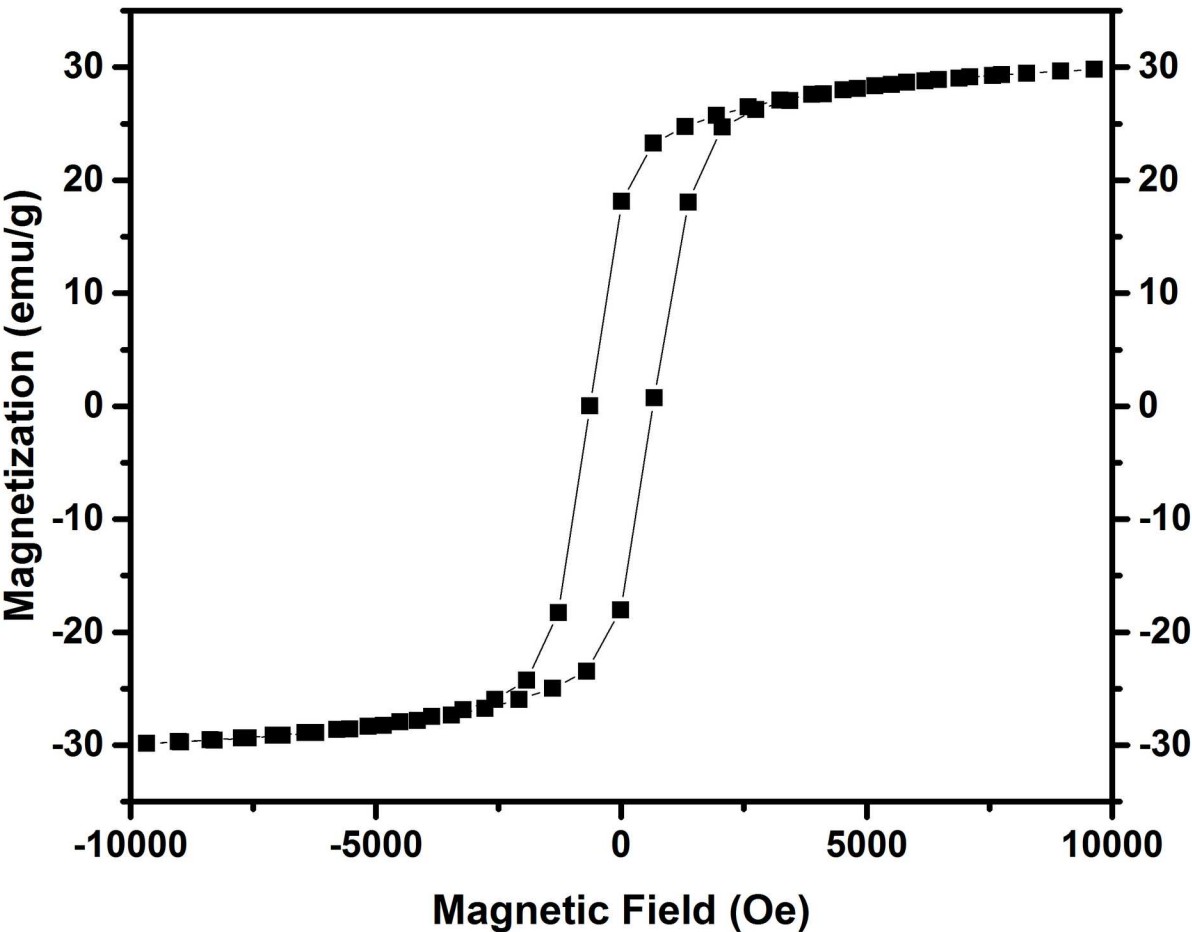

**Fig 6. Hysteresis loop of cerium doped $N_iZ_n$ ferrite, heated at 700˚C during four hours.**

Where 'Ms' is the saturation Magnetization and'$\gamma$' is the gyromagnetic ratio = $2.8 MHz/Oe$. Cerium doped $N_iZ_n$ ferrite sample is suits the sub 6 GHz frequency range. Initial permeability, Y-K angle, anisotropy constant and magnetic moment are calculated by using the following relations [29].

$$Anisotropy \ Constant \ (K_a) = {}^{H_c \times M_s}/_{0.96} \qquad (6)$$

$$Initial \ Permeability \ (\mu_i) = {}^{M_s^2 \times D}/_{ka} \qquad (7)$$

$$nB = (6+x)\cos \propto_{Y-K} - 5(1-x) \qquad (8)$$

**Table 2. Magnetic parameters of $Ni_{0.5}Zn_{0.5}Ce_{0.02}Fe_{1.98}O_4$ nanoferrite.**

| Sample | Saturation Magnetization Ms (emu/g) | Remanence Magnitization Mr (emu/g) | Mr/Ms | Coercivity Hc(G) | $\mu_i$ | Y-K angle (α) | Anisotropy Constant (erg/$cm^3$) | Magnetic Moment (nB) |
|---|---|---|---|---|---|---|---|---|
| NiZn-Ce | 30.15 | 23.55 | 0.6339 | 599.30 | 1.944 | 30.401 | 23,190.71 | 0.28 |

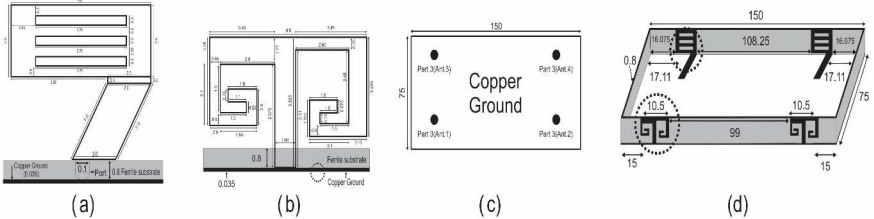

**Fig 7. MIMO antenna model.** (a) window grill component (b) double inverted spiral component (c) grounded Copper d) comprehensive MIMO antenna display.

Where 'Ms' is the saturation magnetization and 'nB' is the experimental magnetic moment for the prepared nano ferrite sample. Table 2 shows the Y-K angle, Anisotropy constant, magnetic moment, saturation, and remanence magnetization and their ratio, coercivity, and initial permeability. These endearing qualities suggest Cerium doped $N_iZ_n$ ferrite with asynthetic formula $Ni_{0.5}Zn_{0.5}Ce_{0.02}Fe_{1.98}O_4$ as a substrate in an antenna for operating at frequency in the C band.

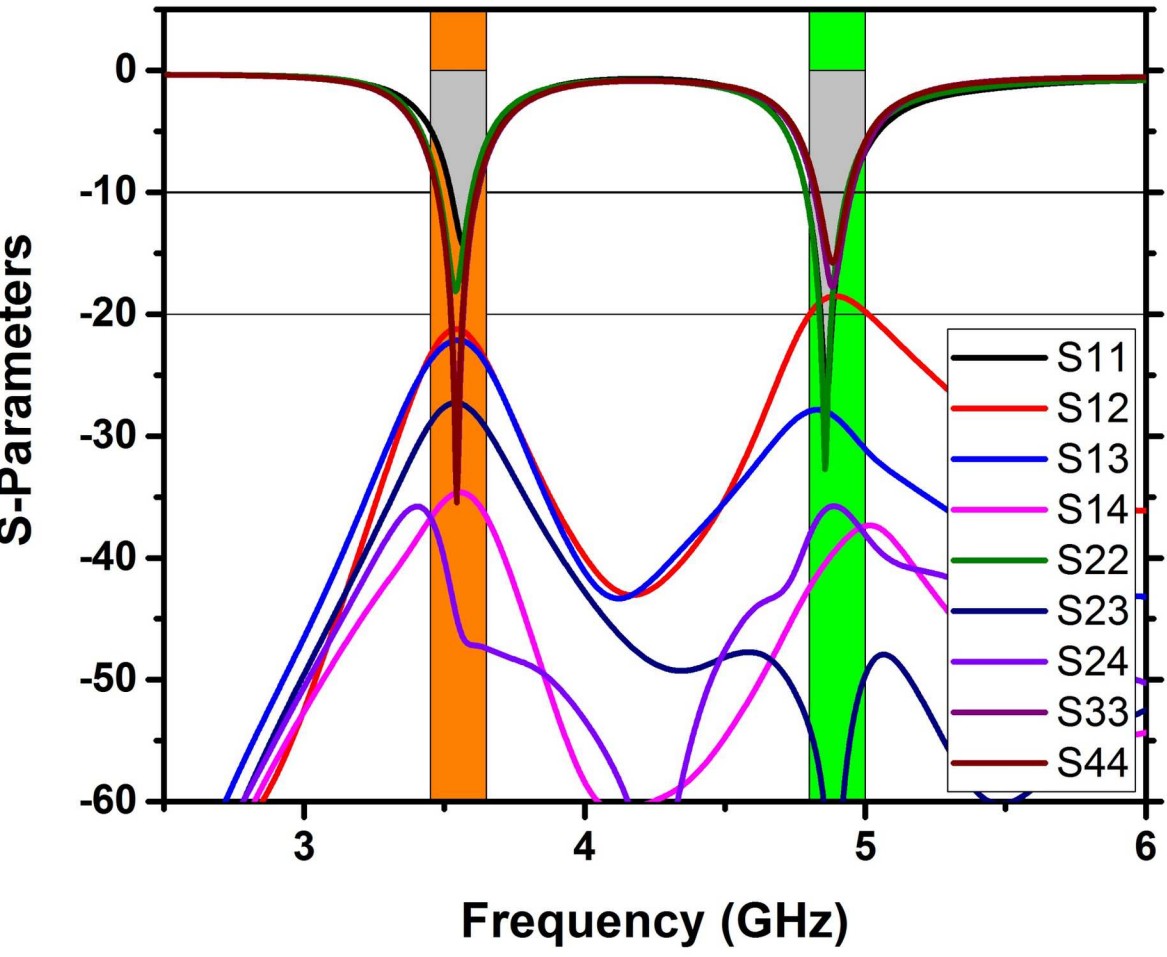

**Fig 8. FEM technique utilized to calculate the S-parameters of 4x4 MIMO antennas, constructed on a ferrite.**

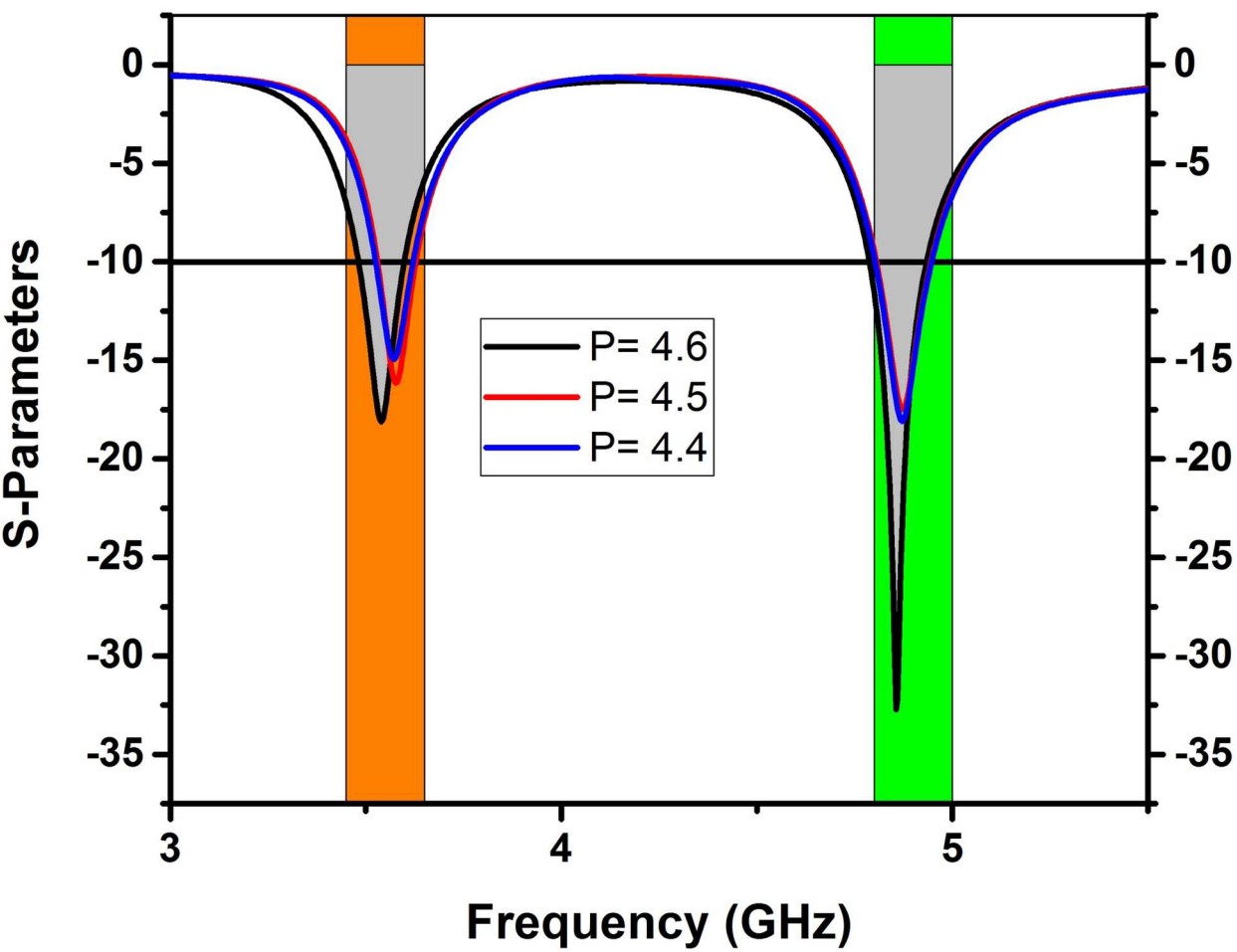

**Fig 9. Effect of change in parameter 'P' on S-parameters.**

When doped into nickel-zinc ferrites, Cerium acts as a grain growth inhibitor. It impedes the movement of grain boundaries during the crystallization process, leading to smaller crystallite sizes. Cerium is known for its mixed valency, which means it can exist in both +3 and +4 oxidation states. This mixed valency can introduce additional magnetic moments in the system and modify the overall magnetic behavior of the material depending on the concentration of Cerium and its distribution in the lattice [30].

Furthermore, rare-earth ion $C_{e^{3+}}$ (1.14 A˚) has bigger ionic radii as compared to $N_{i^{2+}}$, $Z_{n^{2+}}$ nd $F_{e^{3+}}$ ions. The variations in the magnetic parameters are related to ionic radii and magnetic moments of the doping agents such as rare-earth $C_{e^{3+}}$ cations. However, the magnetic moments are noticed to be increased with the rare-earth doping which is the combined effect of the magnetic moments of all the metals cations. Bohr magneton, anisotropy constant, initial permeability, and Y-K angles as depicted in Table 2 confirm the tailoring of magnetic properties due to the incorporation of rare-earth $C_{e^{3+}}$ cation in $N_iZ_n$ ferrite.

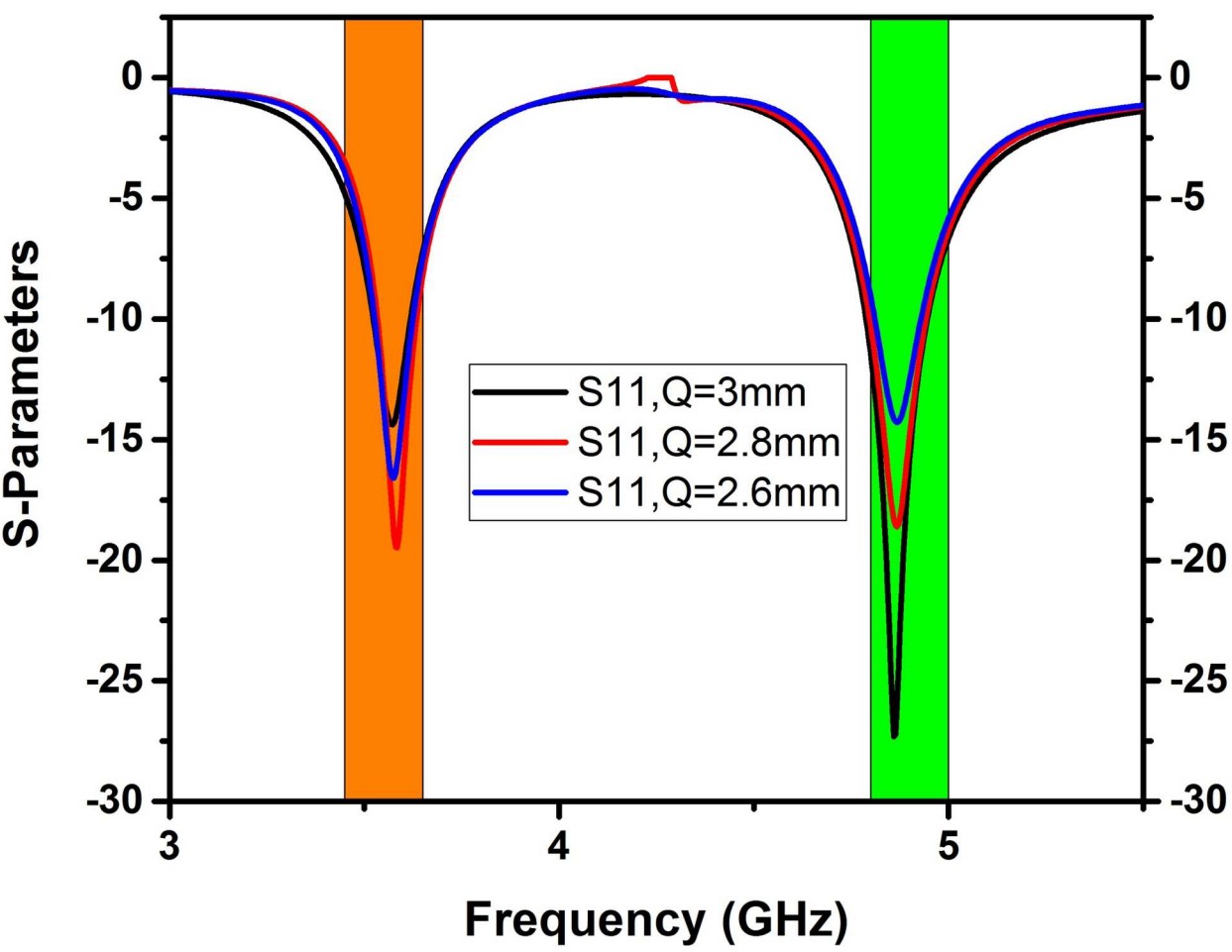

**Fig 10. Effect of changing the parameter 'Q' on the S-parameters.**

## Investigations of dual band MIMO antenna with nano-ferrite substrate

Fig 7 illustrates the configuration and layout of the dual-band MIMO antenna cluster, similar to that of contemporary cell phones with flat and perpendicular substrates. The level substrate is $75 \times 150 \times 0.8$ mm$^3$, whereas the upward substrate is $0.8 \times 150 \times 6$ mm$^3$. Copper-tempered material 0.035mm thick is used for the printed antenna and ground components. The intended antenna has been modeled using the microwave studio suite CST—Microwave Studio. The window grill-molded antenna component is linked to the miniature strip feed line for connectivity. In contrast, the ground-associated inverted spiral-modeled antenna component is positioned on the outer side of the upward-facing substrate. The 50-ohm coaxial port powers the tiny strip feed line.

## Scattering parameters

A double-band 4x4 MIMO antenna with a Cerium doped $N_iZ_n$ ferrite substrate has been examined with the help of the FEM tool of CST software with open boundary conditions [31], implemented on a tetrahedral mesh. Fig 7 displays the S-parameters determined by FEM,

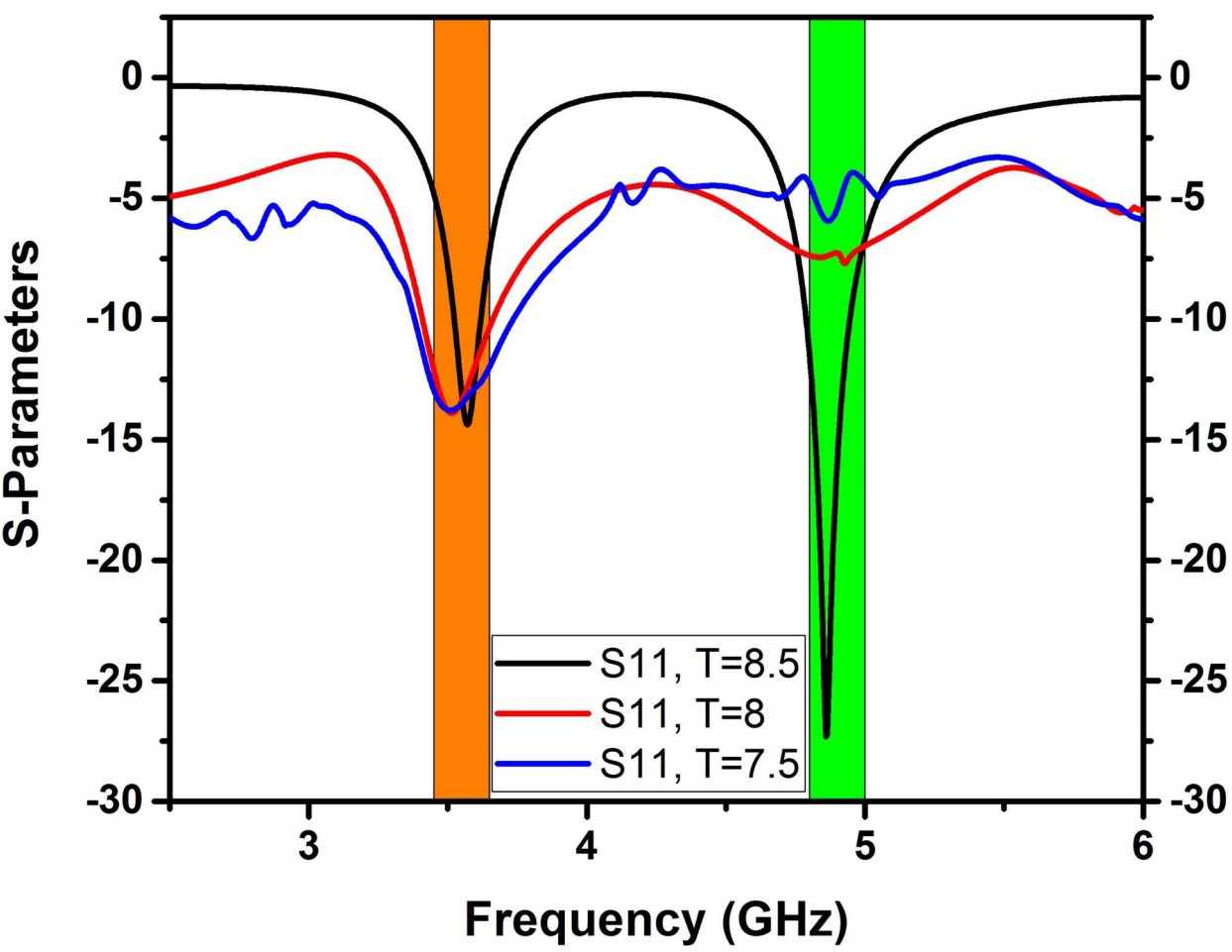

**Fig 11. Effect of changing parameter T on S parameters.**

revealing excellent impedance matching (below -10 dB) for all reflection coefficients ($S_{11}$, $S_{22}$, $S_{33}$, $S_{44}$) at two frequencies, with transmission coefficients indicating minimal isolation of 23 dB at 3.5 GHz and 19.7 dB at 4.8 GHz. Transmission coefficients $S_{21} = S_{12}$, $S_{31} = S_{13}$, and $S_{41} = S_{14}$ were analyzed to account for the symmetry in the calculation. Due to the significant separation between antenna components and the Cerium-doped $N_iZ_n$ ferrite substrate, excellent isolation between adjacent antenna elements is attained without decoupling technology in both frequency bands. Figs 8–11 describe the parametric analysis of the design with the variation of design parameters.

## Envelope Correlation Coefficient (ECC)

It is employed to judge the diversity of a MIMO antenna array. ECC = 0 indicates complete decoupling of the antennas, while ECC = 1 signifies that they are short-circuited. According to Fig 12, the ECC value appears to be 0.00185 in frequency bands of 3.5 and 4.8 GHz, indicating that all MIMO antennas are effectively decoupled.

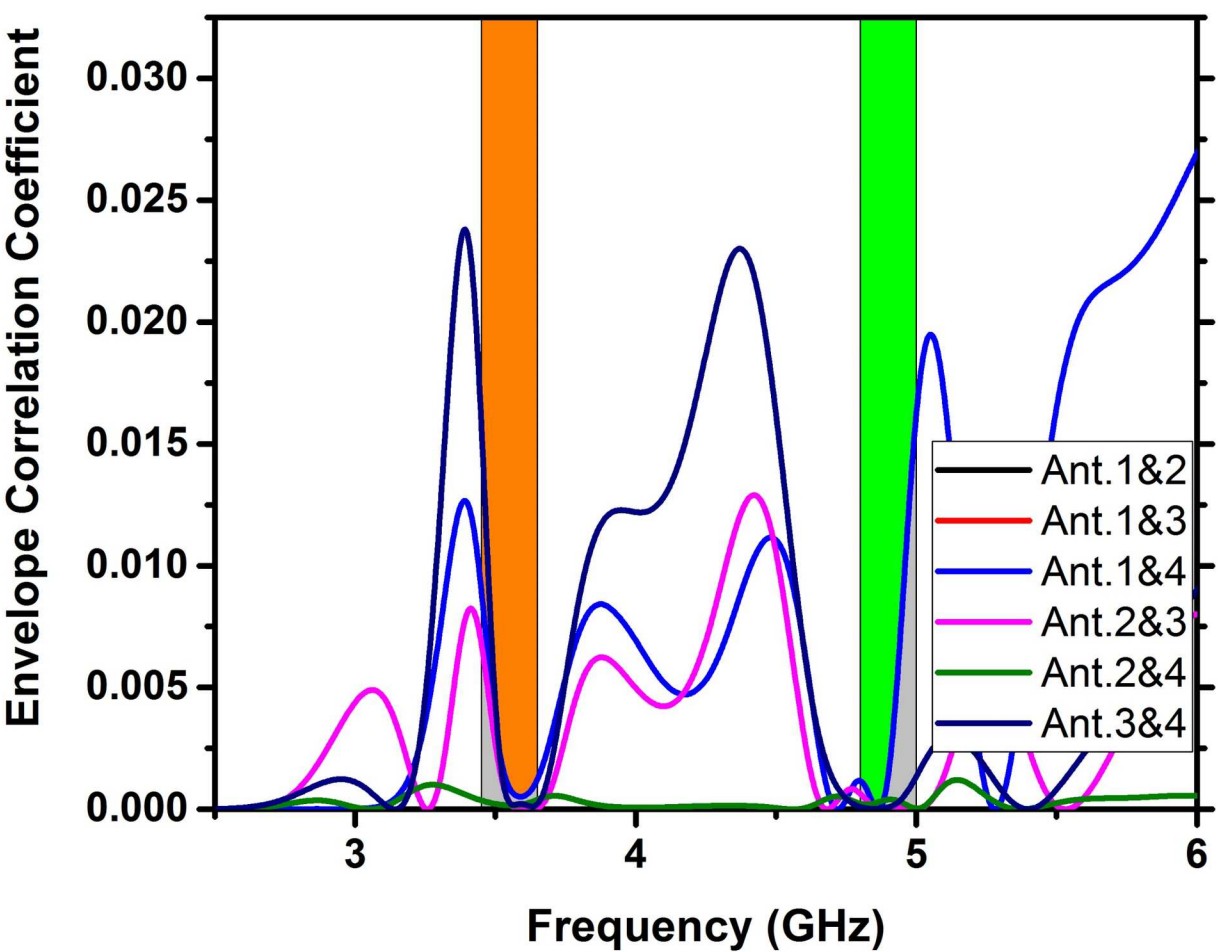

**Fig 12. The ECC value among the antennas of the 4x4 MIMO configurations.**

## CCL—Loss of channel capacity

The loss in the amount of information conveyed across a channel causes a reduction in channel capacity. The CCL can be manually determined using the relations provided below [32].

$$C_{loss} = -Log_2 \det(\varphi^R). \tag{9}$$

$$\varphi^R = \begin{bmatrix} \rho_{11} & \rho_{12} \\ \rho_{21} & \rho_{22} \end{bmatrix} \tag{10}$$

$$\rho_{ii} = 1 - (|S_{ii}|^2 + |S_{ij}|^2) \tag{11}$$

$$\rho_{ij} = -|S_{ii}^* S_{ij} + S_{ji}^* S_{ij}| \tag{12}$$

The CCL is a crucial parameter for determining how well a MIMO antenna performs.

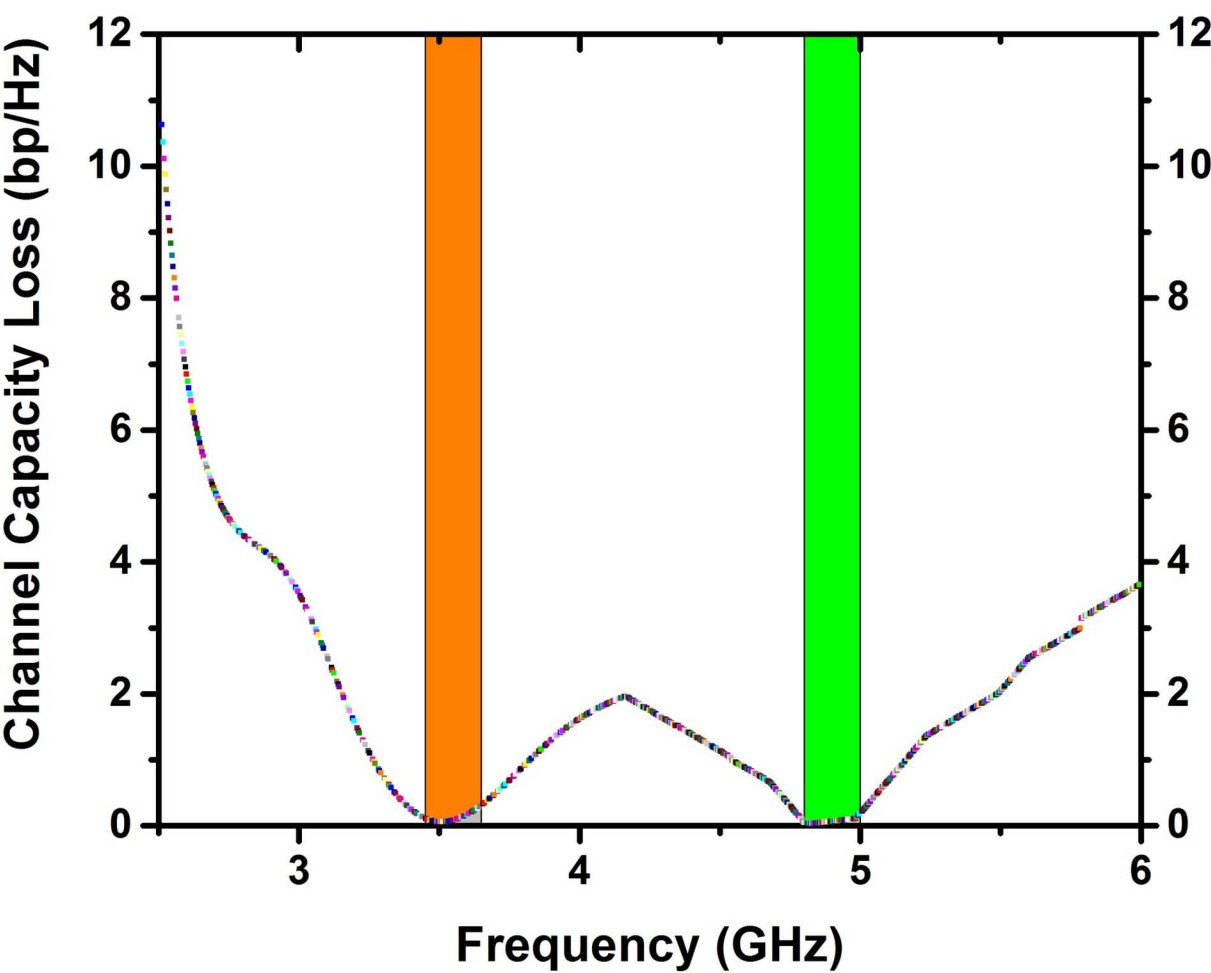

**Fig 13. CCL analysis of 4x4 MIMO antenna.**

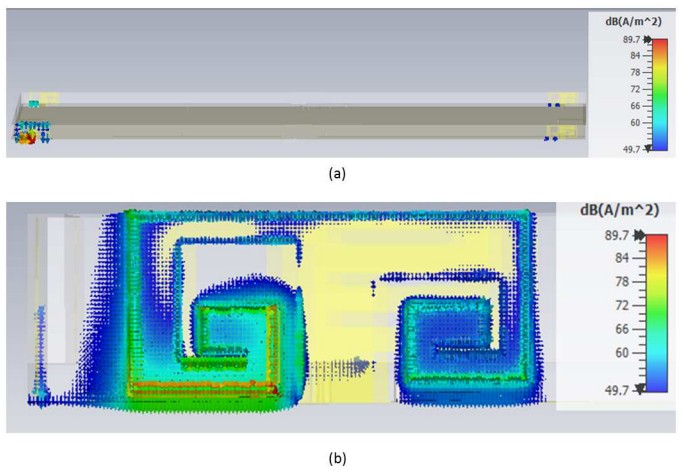

**Fig 14. Surface current distribution with cerium doped $N_iZ_n$ ferrite substrate.**

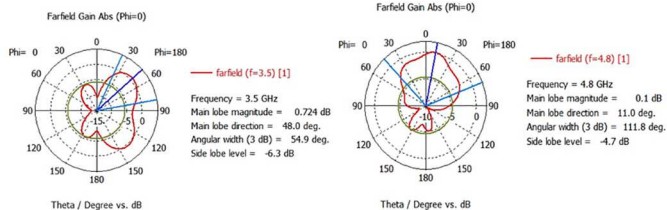

**Fig 15. Antenna 1, 2D radiation pattern with ferrite substrate.**

Fig 13 demonstrate that the CCL value in the frequency ranges of (3.45–3.65 GHz) & (4.80–5.0 GHz) is less than 0.4bps/Hz, which satisfies the necessary threshold level. Consequently, the suggested design of the antenna exhibits exceptional performance in terms of gain diversity.

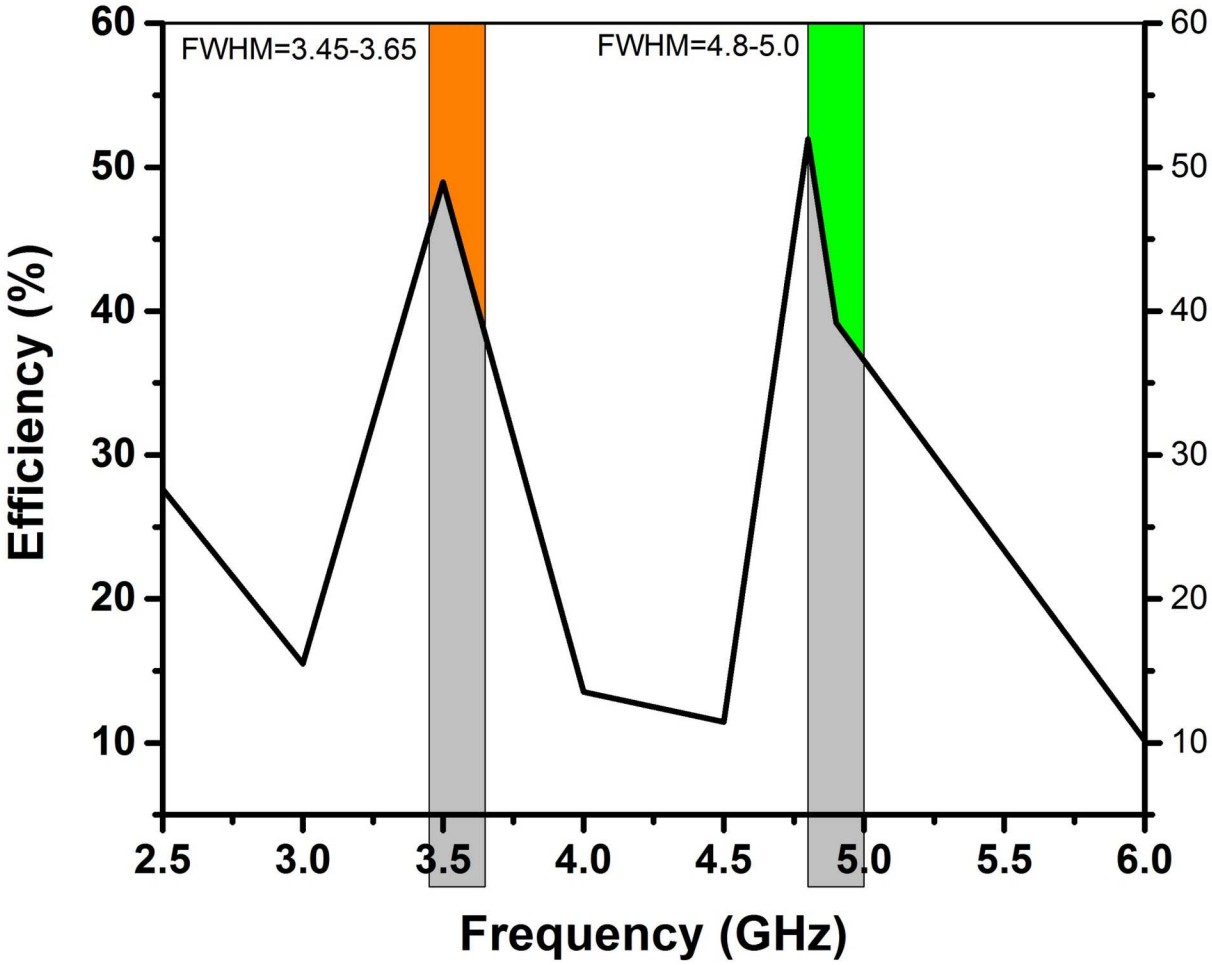

**Fig 16. Antenna 1, radiation efficiency graph with ferrite as substrate.**

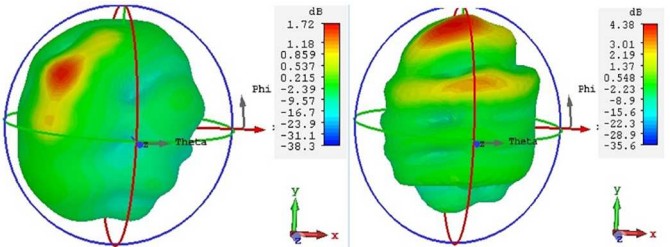

**Fig 17. Antenna 1, gain at 3.5 and 4.8GHz.**

## Surface current analysis

The surface current on metallic antennas is a real electric flow triggered by an applied electromagnetic field [33]. The distribution of surface current on each of the four antennas is shown in Fig 14 when antenna- one is turned on and the other ports are terminated. The current spread of the other three antennas is minuscule, but Antenna 1 has the highest current flow,

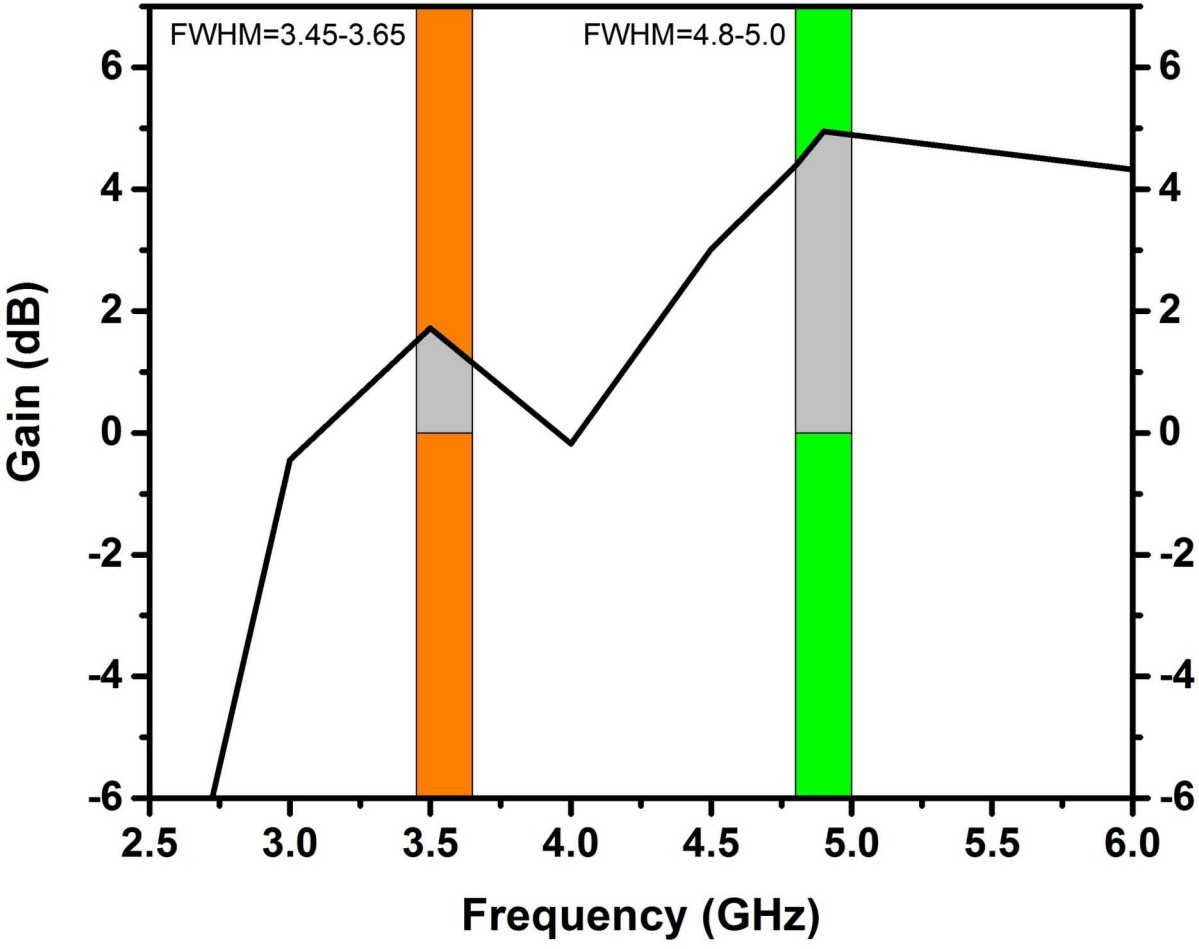

**Fig 18. The gain plot of Antenna 1 utilizing the ferrite substrate.**

**Table 3. Comparative analysis among recent MIMO antennas.**

| Ref. | BW (GHz) | Isolation (dB) | ECC | Gain (dBi) | Antenna Size ($mm^2$) | Techniques Of Isolation | Antenna Array (MIMO) |
|---|---|---|---|---|---|---|---|
| [32] | 3.4–3.6 (-6dB) 4.8–5.0 (-6dB) | 12 | < 0.02 | 4.7 & 5.0 | 3.9×17.7 | Diagonal & Linear organization | 4×4 |
| [13] | 0.72–0.74 (-10dB) | 16.4 | < 0.02 | -8.83 | 14 × 7 | $Ni_{0.5}Mn_{0.2}Co_{0.07}Fe_{2.23}O_4$ ferrite substrate | 2×2 |
| [33] | 3.4–3.6 (-10dB) 4.8–5.0 (-10dB) | 16.5 | < 0.022 | 4.2 & 4.3 | 14.9 ×7 | Distance Optimization | 4×4 |
| [34] | 3.3–3.84 (-10dB) 4.61–5.91 (-10dB) | 15 | <0.02 | 6 & 6 | 14.9 × 7 | Distance Optimization | 4×4 |
| [14] | 3.2–3.65 (-10dB) 5.4–6.0 (-10dB) | 23 & 19 | < 0.002 | 2.2 & 3.6 | 10.5 × 5.535 | $Ni_{0.5}Zn_{0.5}Tb_{0.02}Fe_{1.98}O_4$ ferrite substrate | 4×4 |
| This Study | 3.45–3.65 (-10dB) 4.8–5.0 (-10dB) | 22 & 19 | < 0.002 | 1.89 & 4.38 | 9.5 × 5.535 | $Ni_{0.5}Zn_{0.5}Ce_{0.02}Fe_{1.98}O_4$ ferrite substrate | 4×4 |

indicating more excellent isolation between the antenna parts. For simplicity, the results of antenna 1 in 4×4 MIMO antenna systems are depicted here.

## Radiation pattern

Overall, the radiation performance of an antenna depends on its design and the frequency of operation. Fig 15 describes how antenna 1 radiates energy in different directions in a polar plot resonating at the frequency bands of 3.5 and 4.8 GHz at an angle of Ø = 0 and Ø = 180. Similarly Fig 16 depicts the radiation efficiency of antenna 1.

The gain of an antenna is a measure of its capability to concentrate and direct electromagnetic energy in a particular direction, in contrast to an ideal isotropic radiator that radiates uniformly in all directions [34]. As depicted in Figs 17 and 18, the gain of antenna 1 is 1.89dBi at 3.5GHz and 4.38 dBi at 4.8GHz. Table 3 displays the performance analysis of recently developed MIMO antennas with the proposed design. The table demonstrates that the suggested antenna design, utilizing ferrite as a substrate, showcases superior isolation, bandwidth and gain performance as compared to the recent MIMO antenna designs [13, 14, 35–37].

## Conclusions

Because of its exceptional magnetic and morphological properties, a double-band 4×4 MIMO antenna was built on top of a $Ni_{0.5}Zn_{0.5}Ce_{0.02}Fe_{1.98}O_4$ as ferrite substrate. The resulting ferrite substrate is an intriguing material for upcoming multiband MIMO antenna designs. Given that it resonates at 3.5 and 4.8 GHz with appropriate bandwidths of 3.45–3.65 GHz and 4.80–5.0 GHz, this system is ideal for 5G devices. In both frequency ranges, the CCL and ECC values are below 0.4 bps/Hz and 0.002, respectively. Despite the absence of any decoupling structure, the system provides significant isolation, surpassing 22 dB in the lower-frequency band and

19.0 dB in the upper-frequency band. Hence the proposed Cerium-doped $N_iZ_n$ ferrite materials may be utilized as an antenna substrate in high-frequency devices.

## Supporting information

**S1 File.**
(ZIP)

## Author Contributions

**Conceptualization:** Abdul Aziz.

**Data curation:** Raees Muhammad Asif.

**Formal analysis:** Muhammad Azhar Khan.

**Investigation:** Majid Niaz Akhtar, Muhammad Nawaz Abbasi.

**Project administration:** Muhammad Nawaz Abbasi.

**Resources:** Muhammad Azhar Khan.

**Software:** Raees Muhammad Asif.

**Supervision:** Abdul Aziz, Majid Niaz Akhtar.

**Visualization:** Hafiz Abdul Muqeet.

**Writing – original draft:** Raees Muhammad Asif.

**Writing – review & editing:** Hafiz Abdul Muqeet.

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
