## [Editor Report · Decision Letter 0]

8 Jan 2024

PONE-D-24-00103Synthesis and Characterization of Cerium doped  Nano Ferrites as Substrate Material for Multi Band MIMO AntennaPLOS ONE

Dear Dr. Asif,

Thank you for submitting your manuscript to PLOS ONE. After careful consideration, we feel that it has merit but does not fully meet PLOS ONE’s publication criteria as it currently stands. Therefore, we invite you to submit a revised version of the manuscript that addresses the points raised during the review process.

We look forward to receiving your revised manuscript.

Kind regards,

Anjana Sharma

Academic Editor

PLOS ONE

Journal Requirements:

6. Please upload a copy of Figures 1-18, to which you refer in your text on pages 9-13. If the figure is no longer to be included as part of the submission please remove all reference to it within the text.

Additional Editor Comments:

There is no information regarding why crystallite size decreased with cerium composition. What is the effect of magnetic properties by doping cerium.

---

## [Author Response · Author response to Decision Letter 0]

2 Mar 2024

The response letter to reviewer and editor comments is attached by name of "Rebuttal Letter for PLOS ONE Journal".

---

## [Decision Letter · Decision Letter 1]

12 Mar 2024

PONE-D-24-00103R1Synthesis and Characterization of Cerium doped  Nano Ferrites as Substrate Material for Multi Band MIMO AntennaPLOS ONE

Dear Dr. Asif,

Thank you for submitting your manuscript to PLOS ONE. After careful consideration, we feel that it has merit but does not fully meet PLOS ONE’s publication criteria as it currently stands. Therefore, we invite you to submit a revised version of the manuscript that addresses the points raised during the review process.

We look forward to receiving your revised manuscript.

Kind regards,

Anjana Sharma

Academic Editor

PLOS ONE

Reviewers' comments:

Reviewer's Responses to Questions

**Comments to the Author**

1. If the authors have adequately addressed your comments raised in a previous round of review and you feel that this manuscript is now acceptable for publication, you may indicate that here to bypass the “Comments to the Author” section, enter your conflict of interest statement in the “Confidential to Editor” section, and submit your "Accept" recommendation.

Reviewer #1: (No Response)

Reviewer #2: (No Response)

2. Is the manuscript technically sound, and do the data support the conclusions?

Reviewer #1: Yes

Reviewer #2: Yes

3. Has the statistical analysis been performed appropriately and rigorously? 

Reviewer #1: N/A

Reviewer #2: I Don't Know

4. Have the authors made all data underlying the findings in their manuscript fully available?

Reviewer #1: Yes

Reviewer #2: Yes

5. Is the manuscript presented in an intelligible fashion and written in standard English?

Reviewer #1: No

Reviewer #2: No

6. Review Comments to the Author

Reviewer #1: The manuscript entitled, “Synthesis and Characterization of Cerium doped Nano Ferrites as Substrate Material for Multi Band MIMO Antenna” showcases structural, microstructural, dielectric, and magnetic properties for antenna application. I have attached the remarks for your reference.

Reviewer #2: The manuscript requires major revision and I have attached the list of remarks on the manuscript for amendment in the manuscript.

7. PLOS authors have the option to publish the peer review history of their article (what does this mean?). If published, this will include your full peer review and any attached files.

Reviewer #1: No

Reviewer #2: No

---

## [Author Response · Author response to Decision Letter 1]

18 Apr 2024

The response letter addressing the reviewers' comments has been uploaded for your review.

---

## [Decision Letter · Decision Letter 2]

23 May 2024

Synthesis and Characterization of Cerium doped Ni Zn Nano Ferrites as Substrate Material for Multi Band MIMO Antenna

PONE-D-24-00103R2

Dear Dr. Asif,

We’re pleased to inform you that your manuscript has been judged scientifically suitable for publication and will be formally accepted for publication once it meets all outstanding technical requirements.

Kind regards,

Anjana Sharma

Academic Editor

PLOS ONE

Reviewers' comments:

Reviewer#1: Accept 

Reviewer#2: Accept

---

## [Editor Report · Acceptance letter]

28 May 2024

PONE-D-24-00103R2 

PLOS ONE

Dear Dr. Asif, 

I'm pleased to inform you that your manuscript has been deemed suitable for publication in PLOS ONE. Congratulations! Your manuscript is now being handed over to our production team.

Kind regards, 

on behalf of

Dr. Anjana Sharma 

Academic Editor

PLOS ONE